# Overview of Polyamines as Nutrients for Human Healthy Long Life and Effect of Increased Polyamine Intake on DNA Methylation

**DOI:** 10.3390/cells11010164

**Published:** 2022-01-04

**Authors:** Kuniyasu Soda

**Affiliations:** Department Cardiovascular Institute for Medical Research, Saitama Medical Center, Jichi Medical University, 1-847, Amanuma, Saitama-City 330-0834, Saitama, Japan; soda@jichi.ac.jp

**Keywords:** polyamine, spermine, spermidine, methylation, DNA, lymphocyte function-associated antigen 1 (LFA-1), LFA-1 promoter (ITGAL), DNA methyltransferases (DNMT)

## Abstract

Polyamines, spermidine and spermine, are synthesized in every living cell and are therefore contained in foods, especially in those that are thought to contribute to health and longevity. They have many physiological activities similar to those of antioxidant and anti-inflammatory substances such as polyphenols. These include antioxidant and anti-inflammatory properties, cell and gene protection, and autophagy activation. We have first reported that increased polyamine intake (spermidine much more so than spermine) over a long period increased blood spermine levels and inhibited aging-associated pathologies and pro-inflammatory status in humans and mice and extended life span of mice. However, it is unlikely that the life-extending effect of polyamines is exerted by the same bioactivity as polyphenols because most studies using polyphenols and antioxidants have failed to demonstrate their life-extending effects. Recent investigations revealed that aging-associated pathologies and lifespan are closely associated with DNA methylation, a regulatory mechanism of gene expression. There is a close relationship between polyamine metabolism and DNA methylation. We have shown that the changes in polyamine metabolism affect the concentrations of substances and enzyme activities involved in DNA methylation. I consider that the increased capability of regulation of DNA methylation by spermine is a key of healthy long life of humans.

## 1. Introduction

Biological aging or senescence is associated with declines in physiological function and altered structural changes. The elderly become increasingly susceptibile to aging-associated pathologies such as sarcopenia, frailty, decreases in higher brain function such as decreased cognitive impairment, cardiovascular disease, metabolic diseases, neoplastic diseases, and neurodegenerative diseases. Epidemiological analyses and interventional trials have shown that, among many life-style factors, the differences in food preferences and dietary patterns contribute to the inhibition of aging-associated diseases and senescence. Among them, what has been carefully examined is that the relationship between increased consumption of soybeans and decreases in the incidence of cardiovascular diseases (CVDs) [1,2] and malignancies such as breast [3,4,5] and colon cancer [6,7,8,9], or a Mediterranean diet and increased vegetable intake are associated with a decreased incidence of lifestyle-related diseases, such as CVDs [10,11,12] and breast and colon cancer [13,14,15,16]. These findings indicate that ingredients contained in these foods play an important role in the inhibition of aging-associated pathologies.

Among these substances, antioxidant polyphenols were considered important candidates for extending healthy lifespans. Examples include isoflavones, found at high levels in soybeans, and resveratrol, which is prevalent in the Mediterranean diet. The molecules have many biological activities that counteract the pathogenesis of aging-associated pathologies. For example, they have antioxidant and anti-inflammatory properties [17,18,19,20,21,22], protect cells and genes from harmful stimuli [18,23], increase sirtuin expression, and induce autophagy [17,18,24,25,26,27,28,29,30,31]. Early animal experiments and research performed under specific conditions or in a particular animal demonstrated that the increased intake of polyphenols extended lifespans [32,33]. However, evidence from human intervention studies and recent animal experiments is inconsistent and inconclusive because many studies have failed to show any effects on the prevention of aging-associated pathologies and the extension of lifespan [27,34,35,36,37,38,39,40]. Similarly, vitamin E and β-carotene with potent antioxidant and autophagy-induction properties increased, rather than decreased, the incidence of CVDs and related mortality [41,42,43,44,45,46,47,48].

Based on these scientific facts, we considered that substances other than antioxidants contained in these foods are exerting their health and longevity effects. At that time, we found that natural polyamine synthesized in all cells from the lower primitive organisms to humans and contained abundantly in soybeans and the Mediterranean diet have strong anti-inflammatory properties [49]. Polyamines, spermine and spermidine, in food are absorbed from the intestinal tract, and these polyamines are considered one of the most important sources of polyamines in the body. We also examined the relationship between polyamine content and dietary patterns using the food supply database of 49 Western countries from the Food and Agriculture Organization of the United Nations, and we found that the Mediterranean diet is composed of many polyamine-rich food [50,51]. Simultaneously, we first showed that mouse lifespans were increased by the life-time consumption of chow containing synthetic polyamines with an overall polyamine concentration of about three times that in soybeans [52]. Moreover, when mice with no baseline-elevated risk of carcinogenesis or prior treatment with carcinogenic stimuli were reared on three types of chow with different polyamine concentrations and then multiple, moderate doses of a carcinogen were administered, mice that were fed high-polyamine chow had a significantly lower incidence of colon tumors (most of them were cancer) [53]. In addition, we further examined the biological background of life span extension by increased polyamine intake [53,54,55]. Simultaneously, an intervention trial in humans confirmed the same changes in biological markers observed in mouse models and in vitro studies [56]. The current review will introduce the polyamines in general that has been clarified by many previous papers and discuss the biological background of polyamine-induced life span extension of mammals including humans.

## 2. Polyamine

The natural polyamines (spermine and spermidine) and their precursor, putrescine, are ubiquitous low-molecular-weight aliphatic amines, which contain multiple amino groups (-NH_2_). Spermine and spermidine have four and three amino groups, respectively, and molecular weights of approximately 140 and 200 g/mol, respectively. Putrescine, a precursor of polyamine, has two amines and is therefore referred to as a diamine. Polyamines are synthesized within all living cells. In eukaryotes, polyamine synthesis begins with ornithine, which is synthesized through the urea cycle from arginine. The decarboxylation of ornithine catalyzed by ornithine decarboxylase (ODC) is the rate-limiting step in polyamine synthesis. Spermidine and spermine are then synthesized by the sequential addition of aminopropyl groups donated from decarboxylated S-adenosylmethionine (dcSAM), which is converted from *S*-adenosylmethionine (SAM) by the enzymatic activities of adenosylmethionine decarboxylase (AdoMetDC) (Figure 1).

Spermidine/spermine N-(1)-acetyltransferase (SSAT) and N1-acetylpolyamine oxidase (APAO) degrade intracellular spermine and spermidine. SSAT, a highly inducible enzyme, catalyzes the transfer of acetyl groups from acetyl-coenzyme A to the terminal amines of spermine and spermidine. APAO preferentially catalyzes the oxidation of the N1-acetylspermine and N1-acetylspermidine produced by SSAT activity and yields spermidine and putrescine with the release of an aldehyde and hydrogen peroxide. Alternatively, spermine oxidase (SMO) directly convert spermine to spermidine and release an aldehyde and hydrogen peroxide. Additionally, the polyamine transporter located in the cell membrane can transport polyamines across the cell membrane. The cellular levels of polyamines are tightly regulated through their import, export, synthesis, and catabolism (Figure 1).

Polyamines are universally prerequisite for cell growth and differentiation. However, each of them is of different importance to different organisms. In lower organisms such as bacteria and fungi, putrescine is essential for growth, whereas spermine is not present in the cell [57,58,59,60,61]. In yeasts and nematodes, the concentration of spermine is low and is nonessential for growth [62,63]. These indicate that the role of spermine in cell growth, differentiation, and cell function is insignificant in these lower primitive organisms. Spermine is considered more important in highly developed animals. In fact, a decrease in spermine levels due to a deficiency in spermine synthase have dreadful consequences in humans [64].

In addition to intracellular de novo synthesis, cells can take up polyamine from the extracellular space through a polyamine transporter in the cell membrane. The effects of extracellular polyamine on intracellular polyamine concentration are noticeable in cancer patients. Polyamine biosynthesis is upregulated in cancer cells, and therefore, polyamine concentrations are higher in cancer tissues than in normal surrounding tissues [65,66,67]. Circulating blood cells also take up polyamines synthesized in cancer cells; as a result, the blood cell concentrations and urinary excretion of polyamines, especially those of spermidine, are increased in cancer patients [65,67,68]. These levels decrease after tumor eradication and increase after relapse, indicating that polyamines synthesized in any part of the body, e.g., in cancer tissues, are transferred to blood cells [69].

The polyamines are synthesized from arginine. Arginase converts arginine to ornithine, and ornithine decarboxylase (ODC), a rate-limiting enzyme with a short half-life, catalyzes the decarboxylation of ornithine to form putrescine, a polyamine precursor containing two amine groups. ODC is inhibited by antizyme, and antizyme is inhibited by an antizyme inhibitor. S-adenosylmethionine decarboxylase (AdoMetDC) is the second rate-limiting enzyme in polyamine synthesis and is involved in the decarboxylation of s-adenosylmethionine (SAM). Spermidine synthetase and spermine synthase are constitutively expressed aminopropyl transferases that catalyze the transfer of the aminopropyl group from decarboxylated s-adenosylmethionine (dcSAM) to putrescine and spermidine to form spermidine and spermine, respectively. Polyamine catabolism is mediated by the back conversion pathway in which spermine or spermidine are first acetylated by spermine/spermidine N1-acetyltransferase (SSAT) and then oxidized by N1-acetylpolyamine oxidase (APAO) to yield spermidine or putrescine, respectively. Spermine can be directly converted to spermidine via the spermine oxidase (SMO) reaction. Polyamines are transported across the membrane by the polyamine transporter.

Black text indicates the substance name, while spermidine and spermine are shown in green and blue, respectively. Red letters indicate enzyme names. The solid black arrows indicate the metabolic pathway, and the dashed black arrows indicate the transfer of some material from the upstream material. Thick gray T-bars indicate inhibitory activity on the target.

ODC: Ornithine decarboxylase; SSAT: Spermidine/spermine *N*^1^-acetyltransferase; APAO: *N*^1^-acetylpolyamine oxidase; SMO: Spermine oxidase; SAM: S-adenosylmethionine; AdoMetDC: Adenosylmethionine decarboxylase; dcSAM: Decarboxylated S-adenosylmethionine.

## 3. Aging and Polyamine

The activity of ODC, the rate-limiting enzyme in polyamine synthesis, declines with age [70,71,72]. ODC has been well characterized and has had a short half-life and to be stimulated by various factors [71,73]. The properties of spermidine synthase and spermine synthase have not been fully clarified, however, they seem to lack a regulatory or rate-limiting role in polyamine synthesis. The administration of arginine or ornithine stimulates putrescine levels; however, the subsequent synthesis of polyamines is not necessarily stimulated in elderly people or aged animals [72,74,75,76]. These findings indicate that the activities of spermine and spermidine synthases decrease gradually with aging.

From these findings, it appears that the polyamine concentrations decrease with aging, and it has been reported that there is an age-associated decrease in polyamine concentrations. Madeo et al., citing mainly their own papers, stated that spermidine concentrations decline in an age-dependent manner [77]. However, their group did not show any data on aging-associated changes in polyamine concentrations in humans. The age-associated decline in polyamine concentrations described in the title and abstract in the papers indicate a decline during early life (fetal period or developmental period) [78]. This decrease slows down markedly in adulthood, and there is no significant decrease in healthy adult animals or humans. Nishimura et al. found that polyamine concentrations in various tissues and organs were significantly lower in 10- and 26-week-old mice than in 3-week-old mice, but no differences in spermine and spermidine concentrations were observed between 10- and 26-week-old mice, except the skin [79]. Morrison et al. measured polyamine levels in autopsied human brain, and they reported that no statistically significant influence of aging (from 1 day to 103 years old) on either putrescine or spermine levels, and spermidine levels increased markedly from birth, reaching maximal levels up to 40 years of age and maintained up to old age [80]. Similarly, an age-associated increase, but not decrease, in spermidine concentrations was also reported in a few organs and the semen of animals and humans in good health [81,82].

Polyamine concentrations in blood cells reflect polyamine levels in organs and tissues throughout the body. Elworthy and Hitchcock measured red blood cell polyamine concentrations in 117 patients (ranging from 0 to 80 years old) who were largely in good health but had various neurological problems known not to affect polyamine levels and reported no statistically significant age-dependent changes in spermine or spermidine concentrations [83]. Chaisiri et al. also showed no age-associated decline in plasma polyamine concentrations [84]. Our analyses of aging-associated changes in blood polyamine concentrations in human male volunteers showed no age-associated decline in polyamine concentrations [49,56].

Similarly, urinary polyamine excretion, which reflects blood polyamine concentrations, does not change with age during adulthood. van den Berg et al. measured urinary polyamine excretion in 51 healthy volunteers whose ages ranged from 4 days to 77 years and found an age-dependent decrease in urinary excretion of spermidine in terms of creatinine excretion. However, they clearly indicated that the overall age-dependent decline was merely due to the rapid decrease during the first year of life, and it did not occur during adulthood [85]. Yodfat et al. also examined urinary polyamine concentrations in 171 male and 166 female healthy volunteers whose ages ranged from 14 days to 84 years and demonstrated an age-dependent decrease in diamine levels in males but no age-dependent decrease of polyamines (either spermidine or spermine) in either gender [86]. Several reports have shown that the ratio of spermine/spermidine tends to decrease due to the absence of an age-related decrease in spermidine concentration and an age-related decreasing trend in spermine concentration [56,83,87].

While there is no age-associated decline in polyamine concentrations in tissues, organs, blood, and urine of animals and humans, it is pointed out that there are large inter-individual differences in blood polyamine concentrations [49,83]. The exact biological mechanisms underlying the large inter-individual differences in blood polyamine concentrations are not fully understood. However, these large individual differences in polyamine concentration are one aspect that makes the clinical application of polyamines difficult. In cancer patients, polyamine levels are elevated due to the active synthesis of polyamines in cancer cells, and attempts have been made to diagnose the presence of cancer using polyamine levels as an indicator. Due to this large individual difference, it has been difficult to apply polyamine blood levels and urinary polyamine excretion to the diagnosis of neoplastic diseases. These indicate that reports examining polyamine concentrations in few cases are unreliable. Note that due to the large difference in polyamine concentration, the analysis results vary greatly depending on the choice of cases. For example, Pekar et al. showed that impaired cognitive function is associated with low serum spermidine level [88]. In contrast, Sternberg et al. showed that impaired cognitive function is associated with high serum spermidine levels [89].

## 4. The Effect of Dietary Polyamines on the Body Polyamine

In healthy adult animals and humans, the major source of polyamines is thought to be in the digestive tract, i.e., polyamines in food and polyamines synthesized by the intestinal microbiota. Therefore, a factor to create wide inter-individual differences in polyamine concentrations is thought to be the difference in the polyamine amount supplied from the digestive tract. In fact, a decrease in polyamine intake due to a polyamine-deficient diet and a decrease in polyamine synthesis by intestinal bacteria due to the elimination of intestinal microbiota by antimicrobial agents will lead to a gradual decrease in blood polyamine levels [90,91,92]. Conversely, a long-term increase in the polyamine supply from food gradually increases blood polyamine concentrations, especially spermine concentrations, in humans and mice [52,56,93].

Polyamines exist in all living organisms, and thus, foods that comprise various types of organisms and their related substances contain polyamines, though at a wide variety of concentrations. Germ and bran, legumes such as soybeans, vegetables, and shellfish are foods with high polyamine concentrations per calorie, and spermidine is contained much more than spermine in food [79,94,95,96]. The polyamine concentration in a particular food differs depending on the part of the food examined [96,97]. For example, although fish and shellfish are lower in polyamines than beans and vegetables, higher concentrations of polyamines are found in the internal organs and roe of the fish and shellfish. Therefore, personal food preferences and regional dietary patterns affect polyamine intake from food and influence polyamine levels in the body.

Because polyamine homeostasis maintains individual polyamine concentrations, short-term increases in polyamine supply do not change them [52,56,93,98,99]. Schwarz et al. showed that 28 days of augmentation of spermidine supplementation in mice resulted in no change in blood polyamines in mice [99]. Brodal et al. also reported similar results. For 20 days of feeding with experimental chow with different polyamine concentrations, the levels of putrescine, spermine, and spermidine in rat blood remained remarkably constant irrespective of chow [98]. In our animal experiments using mice, chow containing synthetic polyamine of which concentrations are about 3 times higher than those of soybean failed to increase blood spermine and spermidine concentrations for at least 16 weeks [52,93]. Eisenberg et al. postulated that increased spermidine intake supply spermidine to tissues and organs and spermidine provokes biological activities. They showed using few mice that spermidine ingestion increases blood spermidine levels with large individual differences for 16 weeks, but they did not show a study of a large enough number of mice to be able to rule out the effect of large individual differences in blood spermidine levels [100]. We confirmed that spermine (but not spermidine) concentrations in whole blood of mice fed high-polyamine chow for 26 weeks increased significantly, after repeated attempts with more mice. Spermidine concentrations increased in some animals, however, there were no significant changes [53].

Similar findings were observed in human interventional studies. Schwarz et al. showed that a human trial of 3-months of increased oral spermidine supplementation did not change blood spermidine levels at all [99]. A few intervention studies reported the favorable effects of increased spermidine intake on human memory function, however, they did not present individual data concerning the effect of increased spermidine intake on blood polyamine concentrations or a relationship between spermidine intake and changes in blood spermidine levels [101]. As they seemed to finish a large scale of intervention study [102], I am looking forward to the results of how spermidine consumption affected polyamine levels in humans. In our latest study, in which volunteers were asked to eat fermented soybeans containing high levels of polyamines, blood spermine levels gradually rose very slowly. During the first 8 months of the study, spermine levels slightly elevated in the high-polyamine diet group, however, there was no difference between the control group and the high-polyamine diet group. After 12 months of intervention, the blood spermine level in the high-polyamine group increased with a significant difference [56]. The lack of changes in blood polyamine concentration upon an increase in polyamine intake for a short period indicates that the intracellular polyamine concentration is strictly maintained by polyamine homeostasis. In addition, the findings that the concentration of spermine in blood cells increases only when polyamine intake is increased over a long period indicate that the presence of a long-lasting polyamine supply from the digestive tract can affect polyamine homeostasis and alter intracellular spermine concentrations. However, up to the present, it is unlikely that spermidine supplementation affects spermidine levels in the organs, tissues, or blood in animals and humans because of the lack of data from studies with sufficient populations for analysis.

In both our animal experiments and human interventional studies, increased polyamine intake increased blood spermine levels, while spermidine levels did not change, despite both animals and humans taking more spermidine than spermine [53,56]. Several studies indicate the importance of the composition of the intestinal microbiota for synthesis of intestinal polyamines [103]. Matsumoto et al. reported that probiotics administration increased spermine, but not spermidine, concentrations in feces in humans and animals, although probiotics and intestinal microbiota by themselves cannot synthesize spermine [104,105]. Considering these scientific facts, we can consider that the long-term and continuous increase in polyamine (spermidine > spermine) intake increases the supply of spermine, but not of spermidine, from the digestive tract. The role of intestinal environment and microbiota, especially in the composition of the intestinal microbiota, in polyamine synthesis in the intestinal lumen should be further examined.

In our intervention study using a high-polyamine diet, we found that the relationship between increases in polyamine intake and those in blood spermine levels seems not to be simply additive. The continuous intake of the high-polyamine diet increased blood spermine concentration of the subjects, but the increase in polyamine intake was not necessarily reflected in the increase in blood concentration [52,53,56]. The difference in the increase in blood polyamines in response to increased polyamine intake and wide inter-individual differences in blood polyamine concentration reflect differences in the intestinal environment.

## 5. Polyamine Localization in the Body

Polyamines have a binding capacity to DNA, RNA, and various protein molecules, and are implicated in diverse cellular functions such as transcription, RNA modification, protein synthesis, and modulation of enzyme activities. It has been estimated that a high percentage of total polyamines is bound by ionic interactions to nucleic acids, proteins, and other negatively charged molecules in the cell, while the free intracellular concentration of each polyamine is much lower (7–15% of total for spermidine and 2–5% for spermine in tissues and organs) [106,107]. Therefore, most polyamines in circulating blood are present in blood cells. Copper et al. showed that spermidine and spermine concentrations in plasma account for only 1.0% (spermine) to 1.2% (spermidine) of whole blood [108]. When we measure serum or plasma polyamine concentration by HPLC, it is sometimes hard to detect the peak of spermine due to the low levels [109]. When the concentration of spermine is low, HPLC can only depict the peak of spermine as a shaking of the baseline. We consider that it is difficult to determine accurate polyamine concentrations using such unclear peak. It is also important to note that even if a small amount of hemolysis occurs in the blood sample, the polyamines present in the cells leak out and have a significant effect on the polyamine concentration. The reason for measuring whole blood polyamine levels is to accurately measure all the polyamines present or attached to blood cells.

Blood cells circulate in organs and tissues throughout the body. Polyamine concentrations are increased in cancer tissues due to active polyamine synthesis, and blood polyamine concentrations are increased in cancer patients. These indicate that blood polyamine levels reflect polyamine concentrations in some organs and tissues in the body. Conversely, polyamines in blood cells can be passed to cells in tissues and organs, affecting their concentration. The brain is a typical example of an organ where polyamines in the blood cannot be transferred. Polyamines are highly water-soluble and polar compounds and thus their passage across the intact blood–brain barrier (BBB) is poor [110,111]. Polyamines have been noxious to the BBB by several investigators in different pathological states of the brain, including cerebral ischemia [112,113,114,115]. In animal models, there have been reports of the collapse of the BBB, allowing polyamines to enter brain tissue. However, such disruption occurs only in critical situations such as following traumatic brain injury [116] and ischemic injury [117]. Similarly, BBB dysfunction is reported in 14 patients following traumatic brain injury [118]. Several studies have reported that polyamine aggravate structural defects and even lead to membrane and vascular dysfunction after several different pathological conditions [113,119]. Considering these reports, the BBB protects the brain from being damaged by polyamines entering the brain tissue in otherwise normal conditions, and disruption of the BBB indicates that vascular function and the brain tissue is affected by some serious pathology.

## 6. Biological Activities of Polyamines

The biological activities of putrescine differ from those of polyamines, spermine and spermidine. For example, whereas spermine and spermidine have strong anti-inflammatory activities and are absorbed quickly from the intestinal lumen [120,121], putrescine has little to no anti-inflammatory effect and is degraded in the intestinal lumen [121,122]. Both spermidine and spermine have superficially similar biological activities, however, experiments have shown that spermine has greater potency. In cells with normal homeostasis, the influx of polyamines from the extracellular space suppresses ODC activity. Yuan Q et al. showed that putrescine, spermidine, and spermine inhibited ODC activity stimulated by serum to 85, 46, and 0% of control, indicating spermine is the most, and putrescine the least, effective polyamine in regulating ODC activity [123]. The difference in the strength of the suppression of one of the polyamine synthetic enzymes, e.g., ODC, between spermine and spermidine is reflected in the difference in their ability to regulate DNA methylation and the resultant change in the amount of lymphocyte function-associated antigen 1 (LFA-1), a protein involved in immune function that we have noted. In in vitro studies, we and others found that increasing the spermine concentration to about 1.2 times the level that occurs in vivo resulted in significant biological activity [49,123]. However, intracellular concentrations of spermidine had to increase two- to three-fold in human mononuclear blood cells to elicit obvious biological activities, i.e., the suppression of LFA-1 expression, the production of proinflammatory cytokines, and autophagy induction [49,87,121]. A two- to three-fold change in spermidine concentration is rare, except for a few cases in cancer patients. The difference in activity between spermidine and spermine is also observed in the relationship between blood concentration and LFA-1 levels, one of the bioactive targets of polyamine. When the relationship between polyamine concentration and LFA-1 level was examined in healthy volunteers, blood spermine concentration was inversely correlated with LFA-1 expression, while blood spermidine concentration was not [49,56]. Similarly, Saiki et al. reported that spermine is three to four times more capable of inducing autophagy than spermidine, and that the activity of spermine can be confirmed at concentrations in the physiological range [87].

Polyamines have many biological activities that may counteract the pathogenesis of aging-associated pathologies. For instance, they exert anti-inflammatory [49,121,124,125,126,127,128] and antioxidant properties [129,130,131,132,133,134,135,136,137,138]; protect cells and genes from harmful stimuli such as radiation [139,140,141,142,143,144,145,146,147], ultraviolet rays [148,149,150], toxic chemicals [128,151,152,153,154], and other stresses [148,155,156,157]; and they promote autophagy [128,158,159]. Very interestingly, polyphenols and antioxidant vitamins, which are abundant in legumes such as soybeans and in the Mediterranean diet, also have the same bioactivity as polyamines. For example, they have anti-inflammatory and antioxidant properties [17,19,20,21,22], and they protect cells and genes from harmful stimuli [18,23] and activate autophagy [17,25,26,28,29,30,31]. These biological activities were considered to inhibit the development of age-related pathologies. However, despite the large number of research conducted with antioxidants, the majority failed to show any effects on the prevention of aging-associated pathologies and the extension of lifespan of mammals [27,34,35,36,37,39,40] (Figure 2).

Recently, a research group has highlighted the importance of autophagy-mediated bioactivity by spermidine on life span extension [77]. Autophagy is a natural regulatory mechanism within the cells to remove degenerated or dysfunctional components from the cell. If autophagy function is inhibited, these unwanted components accumulate in cells, inhibit cellular homeostasis, and provoke various pathological changes [160]. Animal experiments using organ-specific conditional autophagy-deficient mice have revealed a close relationship between decreased autophagy and aging-associated pathologies [161,162].

Given that the increases in the incidence of aging-associated diseases with aging, we can speculate that autophagy activity decreases with aging [163]. In addition, decreased autophagy with aging has been reported in various studies [161,164,165,166]. However, other investigators have demonstrated that autophagy does not necessarily decline with age [167,168]. Yamamoto et al. even demonstrated substantial upregulation of autophagy in the aged kidney, suggesting compensatory activation of basal autophagy in response to the increased number of unwanted components with aging-associated accumulation [169].

Experiments using several primitive organisms confirm that inactivation of an autophagy gene suppressed the lifespan extension of long-lived mutant worms and slightly shortened the lifespan of wild-type worms [170,171,172,173,174,175,176]. However, the relationship between autophagy and longevity is not straightforward. Other investigators have shown that the inhibition of autophagy genes in several organisms does not always reduce normal lifespan [163,175,177]. Hashimoto et al. have systematically examined the effect of suppression of 14 autophagy genes on life span using RNAi. They reported that the suppression of autophagy genes can extend, but not shorten, the lifespan of Caenorhabditis elegans. These results indicate that autophagy activation is not necessarily beneficial for longevity and functions to shorten, or at least not extend, lifespan in several nematodes and flies [178]. Additionally, it has also been suggested that physiological levels of autophagy promote survival, while inadequate or excessive levels of autophagy promote death [179]. Because there is little scientific evidence indicating that autophagy activation helps extend the life span of mammals and aging is a complex process in highly developed mammals, the simple relationship between autophagy and aging observed in certain primitive organisms with short life spans is not straightforward in mammals, especially humans.

From these scientific facts, the biological activities described above are not enough to achieve the life span extension and inhibit the progression of senescence in mammals, especially humans, even if the mechanism and/or the pathway by which polyamines and polyphenols elicit these biological activities are different (Figure 2).

Both polyamines (spermidine and spermine) and antioxidants such as polyphenols and antioxidant vitamins have anti-inflammatory properties, antioxidant properties, and protect cells and genes from harmful stimuli and activate autophagy. Despite the vast amount of research on antioxidants, most of the studies have failed to show any benefit in preventing age-related conditions or extending lifespan. Therefore, the biological activities described in the figure are not enough to achieve life span extension and inhibition of the progression of senescence, especially of mammals, even if the mechanism and/or the pathway by which polyamines and polyphenols elicit these biological activities are different.

References for A circled: [17,18,19,20,21,22,23,25,26,28,29,30,31], References for B circled: [49,121,124,125,126,127,128,129,130,131,132,133,134,135,136,137,138,139,140,141,142,143,144,145,146,147,148,149,150,151,152,153,154,155,156,157,158,159], References for “?”: [22,25,26,27,28,30,31].

## 7. Aging, Proinflammatory Status, and DNA Methylation

Mammals are composed of countless numbers of cells and are made up of a complex interplay of various tissues and organs (nervous system, endocrine function, respiratory system, digestive system, immune functions, etc.). In humans, aging is associated with increased susceptibility to numerous different kinds of pathological conditions such as anemia, decreased kidney function, physical function impairment, sarcopenia, metabolic syndrome, diabetes, impaired cognitive function, neurodegenerative diseases, cardiovascular diseases, and cancer. Therefore, the type of disease they suffer from, the organ where pathology develops, and the progression of the disease affect their lifespan.

When considering that aging is associated with an increased incidence of these various diseases, the changes in the body environment that develop with aging play an important role in triggering these various diseases. Among changes, the increased pro-inflammatory status is considered a typical change associated with aging and triggering aging-associated diseases because inflammation and the resulting increase in oxidative stress have been shown to be involved in many aging-associated diseases [180]. Chronic, low-level elevation of proinflammatory cytokines and chemokines, and the resulting increases in inflammatory biomarkers, are associated with age-related declines in function and increased risks of morbidity and mortality [181]. Although the biological background of aging-associated increase in pro-inflammatory status is not fully clarified, one of the typical changes associated with aging responsible for increased pro-inflammatory status is the aging-associated increase in LFA-1 expression in immune cells. [182,183,184,185].

LFA-1 is an important protein involved in the induction of inflammation. The immune cells are activated when they recognize substances to be eliminated, and inflammation is generally the result of immune cell activation to eliminate harmful pathogens. The first step in this process is the binding of LFA-1 on immune cell membranes to intercellular adhesion molecules on endothelial cells lining the innermost layer of blood vessels. The increase in persistent inflammation associated with aging is likely the result of continuously weak stimulation by originally non-stimulatory degraded cells and substances and the increased response of aged immune cells with increased LFA-1 expression [186]. The activation of immune cells results in the production of proinflammatory cytokines, and aging is accompanied by progressive increases in the blood levels of proinflammatory mediators, including tumor necrosis factor α, interleukin-1, and interleukin-6 [187,188,189,190]. All three of these cytokines inhibit erythropoiesis [191], accelerate muscle wasting [192], induce insulin resistance [193], promote vascular dysfunction [194], and the resulting chronic inflammation provoked by cytokine production has also been linked to neurodegenerative diseases such as Alzheimer’s disease [195] and carcinogenesis [196].

The amount of LFA-1 protein on immune cells’ membrane is regulated by two main mechanisms. One is the intracellular signaling pathways, which result in rapid redistribution of LFA-1 to the cell membrane, regulating LFA-1 levels [197]. Apart from that, the alteration of DNA methylation status of the responsible part of LFA-1 expression, called ITGAL, also regulate LFA-1 expression. DNA methylation is a mechanism to regulate gene expression by modulating methylation in genomic regions that are either distal or proximal to the transcription start site of a gene. It has been reported that increases in LFA-1 protein levels in immune cells with aging are associated with enhanced demethylation of the ITGAL (LFA-1 promoter area) and increased LFA-1 protein levels [49,56,198,199].

A growing number of recent studies have shown a close relationship between aging and DNA methylation [198,200,201]. Aging is associated with enhanced demethylation of DNA in various organs and tissues in several animals and humans [202,203,204]. However, increased methylation (hypermethylation) associated with age has also been reported in other genes [205,206]. The condition in which demethylation and hypermethylation are present in various parts of the entire gene is called the aberrant DNA methylation. The progression of aberrant DNA methylation changes are key regulators of the aging process and contributors to the development of aging-associated diseases [207,208,209,210,211,212,213], including neoplastic growth [214,215,216,217,218] and senescence [219,220,221,222].

DNA methylation is susceptible to various lifestyle and living environments, such as environmental air pollution [223,224,225,226,227], smoking [218,228,229,230,231,232], and excessive alcohol consumption [233,234,235,236]. It has been reported that the changes observed on DNA methylation induced by these environmental factors with negative health effects have similarities to those associated with aging-associated diseases, such as malignant transformation [218,232,237], CVDs [231,238], and the acceleration of senescence [229,230,239]. Contrarily, lifestyle habits considered having favorable consequences for health, such as exercise [240,241,242,243] and dietary restriction for obesity [244], also alter DNA methylation status, which is the opposite of the changes observed during aging.

## 8. Polyamine, DNA Methylation, and LFA-1

Age-related genome-wide aberrant DNA methylation status [222,245], enhanced demethylation of the LFA-1 promoter area [199], and increases in LFA-1 protein levels [49,56,185,198] are accompanied by decreases in ODC [70,246,247] and DNA methyltransferases (DNMTs) activities [248,249,250,251]. There is a close relationship between ODC, a rate-limiting enzyme for polyamine synthesis, and DNMTs involved in DNA methylation. Ornithine and SAM are two crucial substrates for polyamine synthesis. Ornithine is converted to putrescine by ODC, and SAM is converted to dcSAM by AdoMetDC. Putrescine receives the aminopropyl group from dcSAM and is sequentially synthesized into spermidine and spermine. DNA methylation is the conversion of cytosine residues to 5-methylcytosine by the addition of the methyl group to a cytosine residue at the C-5 position. DNMTs regulate methylation of DNA in the presence of SAM, while dcSAM is a strong inhibitor of DNMTs [252]. The activity of DNMT is closely associated with the concentration of SAM [253] and dcSAM [254,255], and with the dcSAM to SAM ratio [252,255] (Figure 3).

The relationship between polyamine metabolism (left side) and gene methylation (right side) is indicated. S-adenosylmethionine (SAM), an amino acid, is a substrate for polyamine synthesis and a donor of methyl groups. During polyamine synthesis, spermidine and spermine synthase require an aminopropyl group from decarboxylated s-adenosylmethionine (dcSAM), which is converted from SAM by the enzymatic action of adenosylmethionine decarboxylase (AdoMetDC). DNA methyltransferases (DNMTs) regulate gene methylation status by receiving a supply of the methyl group from SAM. SAM is essential as a source of methyl groups in gene methylation reactions, and dcSAM is a strong inhibitor of DNMTs.

Black text indicates the substance name, while spermidine and spermine are shown in green and blue, respectively. Red letters indicate enzyme names. The solid black arrows indicate the metabolic pathway, and the dashed black arrows indicate the transfer of some material from the upstream material. The thick gray arrow indicates activity on the target, and thick gray T-bar indicates the inhibitory activity on target.

ODC: Ornithine decarboxylase; SAM: S-adenosylmethionine; AdoMetDC: Adenosylmethionine decarboxylase; dcSAM: Decarboxylated S-adenosylmethionine; DNMT: DNA methyltransferase.

Intracellular concentrations of dcSAM rise in cells when the addition of aminopropyl groups for polyamine synthesis is no longer necessary. This condition is observed when ODC activity is decreased due to the overexpression of antizyme that degrade ODC or treatment with α-D,L-difluoromethylornithine hydrochloride (DFMO), which inhibits ODC activity [252,255,256,257]. An increase in dcSAM induced by inhibition of ODC activity decreases DNMT activity [254,255,258]. We also showed that ODC inhibition by DFMO increased dcSAM concentrations and the dcSAM/SAM ratio and decreased activities of DNMT 1, 3a, and 3b in Jurkat cells [55]. We initially speculated that decreased DNMT activity and decreased donation of methyl groups to cytosine residues would lead to progressive demethylation of the entire genome, but in fact, aberrant methylation of the entire genome was promoted. The decline in DNMT activity induced by the inhibition of polyamine synthesis both increased demethylation in certain areas and increased hypermethylation in other areas, resulting in genome-wide aberrant methylation status [53,259,260,261,262,263]. Simultaneously, polyamine-deprivation enhanced demethylation of ITGAL region and increased LFA-1 protein levels [54]. The changes in methylation status in the ITGAL region were similar to those observed in the entire genome, i.e., aberrant methylation. In other words, some sites in ITGAL were demethylated, some were hypermethylated, and sites important for LFA-1 expression in immune cells were demethylated upon polyamine deprivation, increasing LFA-1 protein levels. The increase in LFA-1 expression associated with aging has been reported in animals and humans [53,182,183,184,185]. In addition, the age-dependent increase in the LFA-1 expression is associated with the age-dependent methylation changes in ITGAL region [53,198,264].

As we have shown in our animal experiments and human intervention studies, increased polyamine intake (spermidine content is much higher than spermine content) elevates blood spermine levels [52,53,56,93]. Many reports have shown that the biological activities of spermine are much stronger than those of spermidine [49,87,98,123]. Therefore, spermine was employed to study its effects on enzyme activities and substance concentrations involved in polyamine synthesis and DNA methylation. In cells with or without suppressed ODC activity by DFMO, spermine supplementation inhibited AdoMetDC activity and decreased dcSAM concentrations with a decreased dcSAM/SAM ratio. Decreasing dcSAM concentration or decreasing dcSAM/SAM ratio by spermine supplementation weakened the inhibitory effect of dcSAM on DNMT3a and 3b and activated these two enzymes (Figure 4), but did not reactivate DNMT1, which was inhibited by polyamine deprivation by DFMO [55]. Similarly, it has been reported that the presence or absence of methyl donors affects the expression of DNMT3a and 3b by other researchers [253]. DNMT3a and 3b are involved in de novo methylation and DNMT1 is involved in the maintenance of gene methylation status. Spermine supplementation reversed the changes in methylation status, i.e., a reversal of increased demethylation, in ITGAL induced by polyamine deprivation, resulting in decreased LFA-1 protein levels [54] (Figure 4).

Increased polyamine intake elevates blood spermine levels and inhibits ODC activity. Increased spermine concentration strongly suppresses AdoMetDC activity, resulting in an increased amount of SAM and reduced amount of dcSAM. Since SAM is a methyl group donor for DNA methylation and dcSAM inhibits the activity of DNMTs, DNMTs are activated. As a result, enhanced aberrant methylation of entire genome and increased demethylation of ITGAL are reversed and regulated.

Black text indicates the substance name, while spermidine and spermine are shown in green and blue, respectively. Red letters indicate enzyme names. The solid black arrows indicate the metabolic pathway, and the dashed black arrows indicate the transfer of the methyl group from SAM. The brown arrows indicate the conditions of enzymatic activities (upward and downward arrows). Upward arrows indicate activation of the enzyme, and downward arrows indicate the inhibition of enzyme activity. Green arrows indicate the change in material quantity and enzymatic activity. The thick gray arrows indicate the stimulus given to the target by the upstream enzyme activity, and the thick gray T-bars indicate the inhibitory activities on the target.

The right figures show the condition and changes in DNA methylation status. The length of the line of black circles with bars indicates the progression of demethylation and hyper-methylation. The upward line indicates the progression of demethylation, and the downward lines indicate the progression of hyper-methylation.

ODC: ornithine decarboxylase; SSAT: Spermidine/spermine *N*^1^-acetyltransferase; APAO: *N*^1^-acetylpolyamine oxidase; SAM: S-adenosylmethionine; AdoMetDC: Adenosylmethionine decarboxylase; dcSAM: Decarboxylated S-adenosylmethionine; DNMT: DNA methyltransferase; ITGAL: gene promoter area that is responsible for the LFA-1 expression.

Spermine can reverse the methylation status of ITGAL in several cells. However, spermine seems not to be able to reverse all the methylation changes associated with aging. In our earlier report, we noticed that the effects of spermine on LFA-1 expression are different from those of aging [49]. And, in our latest intervention study, we observed that the group of cells in which LFA-1 expression increases with age is different from the group of cells in which LFA-1 expression decreases by spermine supplementation, suggesting that the sites and cells in which spermine alters methylation status are different from those affected with aging [56]. Additionally, both abnormal genome-wide methylation and elevated LFA-1 protein levels observed in aged mice fed a normal chow were significantly suppressed, though not completely restored, in mice fed a high-polyamine diet [53]. Based on these results, spermine does not necessarily have effects on chronologically induced methylation changes. However, the regulation of DNA methylation by spermine may inhibit the onset and progression of various pathological changes and lifestyle-related diseases and consequently to slowing down the progression of senescence.

## 9. Possible Role of Polyamine in Cognitive Function

The role of polyamines in cancer progression is well known. Polyamine accelerates neoplastic growth and enhance metastatic spread [69]. However, it is controversial whether polyamines act as an initiator of carcinogenesis in normal cells that do not have existing carcinogenic elements (genetic abnormalities leading to carcinogenesis, exposure to preceding carcinogens and carcinogenic stimuli, etc.). There are several reports suggesting that increased polyamine intake does not increase carcinogenesis but has a suppressive effect on carcinogenesis in healthy individuals [53,265,266]. I have discussed the issues in my previous review [267].

In this section, I will discuss the possible role of polyamine in the inhibition of cognitive decline or the improvement of cognitive function. One of the major risk factors for poor health and shortened life expectancy among the elderly is the incidence and progression of diseases associated with cognitive decline and impaired cognitive function [268,269,270]. Individuals with cognitive impairment with or without definite neurodegenerative diseases have a higher mortality risk than healthy controls [270,271,272,273,274]. Therefore, it would make sense to investigate the possible role of increased polyamine intake, which extends the lifespan, in these aging-associated changes.

The possibility that spermidine is involved in memory function through a mechanism involving a novel memory-related molecule has been reported in insects such as Drosophila [275]. It cannot be completely ruled out the possibility of these mechanisms to function in the learning memory of higher animals such as mammals including humans. However, differences in basic biological background between insect memory and human memory are not known in detail, and the biological activities of spermidine in insects are not similar to its role in humans (described in Section 2. Polyamine). Additionally, because polyamines cannot cross the BBB under otherwise normal conditions, it is not expected that spermidine or spermine supplied from the intestinal lumen enters brain cells and exert their biological activities. Moreover, the role of autophagy activation in memory function and life span extension of mammals, especially humans, is not known (described in Section 6. Biological activities of polyamine). Therefore, I think it is difficult to believe that the effects on cognitive functions confirmed in some insects can be demonstrated in humans.

While polyamines cannot cross the BBB under normal conditions, reports have indicated that BBB dysfunction is associated with the pathogenesis of various neurodegenerative disorders such as Alzheimer’s disease [276,277], Parkinson’s disease [278,279], multiple sclerosis [280,281,282], and amyotrophic lateral sclerosis [283,284], in addition to typical cerebrovascular disorders such as stroke and vascular dementia [285,286]. However, it is incomprehensible how an increase in spermidine, which is so minuscule that it cannot elicit or enhance a physiological function, can exert any physiological activities. The effects of the stimulation of autophagy by the substances on cognitive function are also not clearly defined in humans. Moreover, even when the disruption of the BBB found in patients with cognitive impairment allows spermidine to enter brain tissue, patients with impaired cognitive function already have higher spermidine levels in the brain and blood than normal volunteers. Inoue et al. showed that patients with neurodegenerative diseases such as Alzheimer’s disease have increased spermidine levels in the frontal and parietal lobes of the brain [287]. Although polyamines cannot cross the BBB, their brain concentrations can be reflected in the blood via cerebrospinal fluid [288]. Sternberg et al. showed that serum levels of spermidine in patients with mild cognitive function are higher than those of healthy controls. Additionally, they reported that high spermidine level is associated with impaired cognitive function [89]. Similarly, the mean value of blood SPD levels in the Parkinson’s disease group was 134% higher than those of the controls [289]. Moreover, Graham et al. reported that spermidine plasma levels in patients with mild cognitive function who subsequently converted to Alzheimer’s disease were higher than those who did not [290]. Elevated spermidine levels in neurodegenerative diseases are consistent with reports of decreased spermine/spermidine ratios in patients with such diseases. Saiki et al. reported that spermine and the spermine/spermidine ratio is decreased in patients with Parkinson’s disease and Alzheimer’s disease [87].

There is evidence indicating a close relationship between chronic inflammation and neurodegenerative diseases [195,291,292,293]. Inflammation activates SSAT, an enzyme that breaks down spermine to spermidine and spermidine to putrescine, decreasing polyamine levels (Figure 1 and Figure 5). Increased polyamine degradation and decreased concentrations of spermine and spermidine activate polyamine recycling pathway, i.e., the activation of enzymes for polyamine synthesis. It is unclear how these metabolic changes affect polyamine concentrations, but it can be inferred by their effects on spermine and spermidine concentrations under different pathological conditions. In cancer tissues, inflammation and autonomous, but not stimulation-induced, increases in polyamine synthesis are observed. In cancer tissues, increases in spermidine concentrations are more prominent than spermine [65,67]. Similarly, in patients with neurodegenerative diseases, there are decreased spermine/spermidine ratios or increased spermidine concentrations [87,89,287,290]. These findings further support that inflammation-induced changes in polyamine metabolism are involved in the background of neurodegenerative diseases and cognitive decline. Our latest research also shows that the spermine/spermidine ratio in the blood of elderly people who can live independently is higher, i.e., increased spermidine concentrations relative to spermine, than that of older people who compelled to live in the elderly care facilities for assistance (manuscript is being prepared).

Due to the feedback mechanism, the activation of polyamine degradation by inflammation stimulates the enzymes of polyamine synthesis, and one of these enzymes, AdoMetDC, is also stimulated (Figure 5). Morrison LD et al. reported that autopsied brain of patients suffered Alzheimer’s disease showed an increase in the activity of AdoMetDC [294], also suggesting the involvement of chronic inflammation in the pathogenesis of the disease. AdoMetDC converts SAM to dSAM, and dSAM is used for polyamine synthesis. Thus, increased AdoMetDC activity decreases the SAM available for DNA methylation (Figure 5). Over the long term, this condition results in the progression of aberrant DNA methylation in the entire genome [295].

Age-associated chronic inflammation activates SSAT. SSAT activation enhances spermine degradation and results in decreased spermine concentration. Polyamine synthesis is activated as a compensation for polyamine degradation, resulting in an activation of AdoMetDC. AdoMetDC consumes SAM for polyamine synthesis and results in a decreased supply of methyl groups for DNA methylation. The lack of a methyl group supply results in aberrant methylation of the entire genome and increased demethylation of ITGAL.

Black text indicates the substance name, while spermidine and spermine are shown in green and blue, respectively. Red letters indicate enzyme names. The solid black arrows indicate the metabolic pathway, and the dashed black arrows indicate the transfer of the aminopropyl group from dcSAM. The brown arrows indicate the conditions of enzymatic activities (upward and downward arrows). Upward arrows indicate the activation of the enzyme, and downward arrows indicate the inhibition of enzyme activity. Green arrows indicate the change in enzymatic activity. The thick gray arrows indicate the stimulus given to the target by the upstream substance or enzyme activity, and the thick gray T-bars indicate the inhibitory activities on the target.

The right figures show the condition and changes in DNA methylation status. The length of the line of black circles with bars indicates the progression of demethylation and hyper-methylation. The upward line indicates the progression of demethylation and the downward lines indicate the progression of hyper-methylation.

ODC: ornithine decarboxylase; SSAT: Spermidine/spermine *N*^1^-acetyltransferase; APAO: *N*^1^-acetylpolyamine oxidase; SAM: S-adenosylmethionine; AdoMetDC: Adenosylmethionine decarboxylase; dcSAM: Decarboxylated S-adenosylmethionine; DNMT: DNA methyltransferase; ITGAL: gene promoter area that is responsible for the LFA-1 expression.

Continuously long-term increases in polyamine intake increased only spermine levels in the blood and suppressed aging-associated increases in proinflammatory conditions in humans and mice [53,56], i.e., suppression of LFA-1 protein levels on immune cells, and suppressed aging-associated enhancement of aberrant methylation of the entire genome and extended life span of mice [53]. These changes were associated with increases in blood spermine levels, and the degree of increase in the concentration of spermine was sufficient to make it biologically active in vivo. Increases in blood spermine levels are also sufficient to strongly suppress AdoMetDC activity, resulting in decreased dcSAM concentrations and increased SAM availability (Figure 4). The increase in SAM and the decrease in dcSAM that accompanies increases in spermine concentration activate DNA methyltransferase, which in turn regulates the DNA methylation status of the entire genome of brain tissue, tilting the LFA-1 promoter region toward hypermethylation and leading to a less pro-inflammatory state, i.e., reduced expression of LFA-1.

In both humans and rodents, the global changes in DNA methylation with normal aging are found in various tissues, including the brain [296,297,298,299]. Gene-specific DNA methylation changes are essential for memory formation, neurogenesis, and neuronal plasticity [300,301]. Unlike polyamines, which are difficult to pass through the BBB, SAM and dcSAM can pass through the BBB. It is reasonable to assume that changes in the concentration of substances related to DNA methylation and changes in enzyme activity caused by increased polyamine intake can help improve cognitive dysfunction via the regulation of methylation status and the suppression of proinflammatory condition. Very interestingly, the possible role of SAM supplementation in cognitive function and neuropsychiatric disorders has been discussed [302,303,304]. Our in vitro experiments showed that SAM supplementation reversed polyamine-deficient induced increase in LFA-1 expression, indicating a reversal of aberrant methylation status of the entire genome induced by polyamine deficiency [54]. Spermine supplementation not only reversed polyamine-deficient induced increase in LFA-1 but also further decreased LFA-1 expression, suggesting its potent effect on the regulation of DNA methylation [54].

## 10. Conclusions

We started our experiments to investigate the role of polyamines in extending the lifespan and health of mice, when we first discovered that spermine selectively suppresses LFA-1 expression in immune cells in 2005 [49]. After repeated experiments on mice, we confirmed the effect of increased polyamine intake on life span. After a long period of non-responsive and silent review of a submitted paper, we could publish for the first time the effect of increased polyamine intake on the lifespan of mice [52]. I hope this review will be helpful to scientists who are investigating the health and longevity of mammals, especially humans. I summarized the bioactivities and mechanisms of polyamines contributing to the extension of healthy life span (Figure 6). 

The mechanism by which increased polyamine intake inhibits onset or progression of aging-associated diseases and senescence. Increased polyamine intake elevates blood spermine levels in humans, in spite the fact that many foods contain spermidine much more than spermine. Polyamine binds to the cell membrane, proteins, and genes by electric charge. Polyamine (spermine and spermidine) protects cells and genes from harmful stimuli indicated in red. Spermine inhibits aberrant DNA methylation and regulates DNA methylation status. These biological activities contribute to a healthy longevity.

## Figures and Tables

**Figure 1 cells-11-00164-f001:**
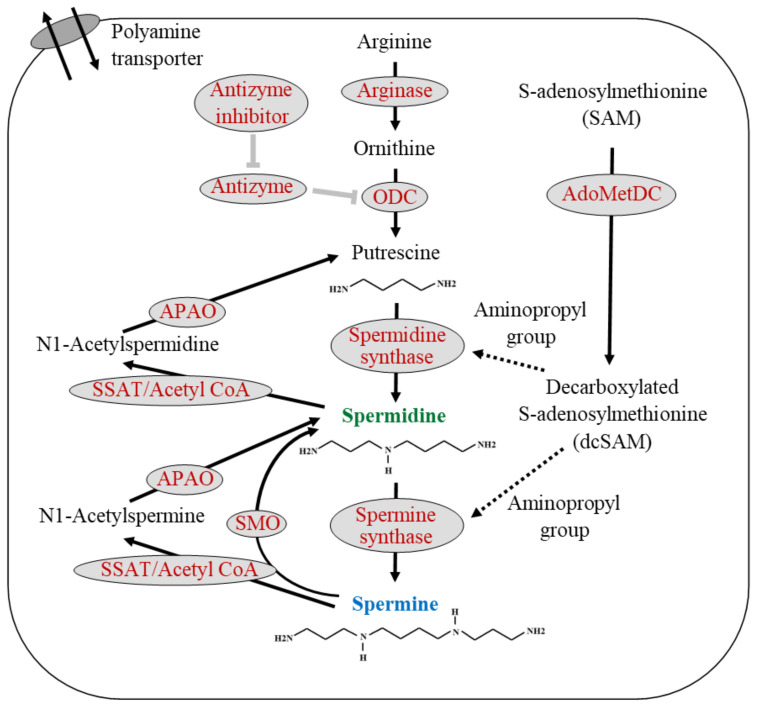
Polyamine biosynthesis, degradation, and transmembrane transport.

**Figure 2 cells-11-00164-f002:**
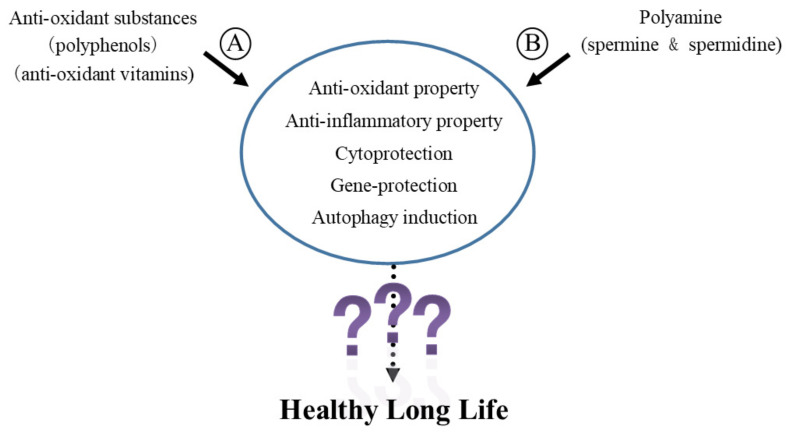
Biological activities of polyamine and polyphenols.

**Figure 3 cells-11-00164-f003:**
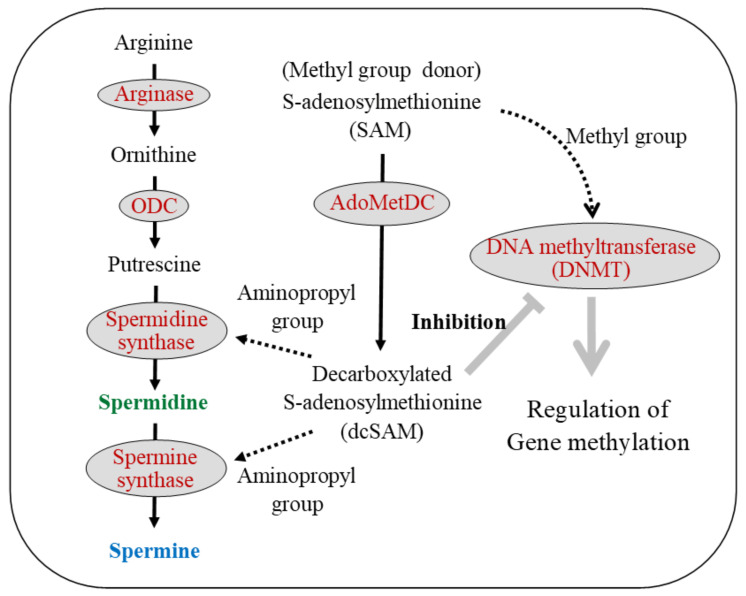
Polyamine metabolism and gene methylation.

**Figure 4 cells-11-00164-f004:**
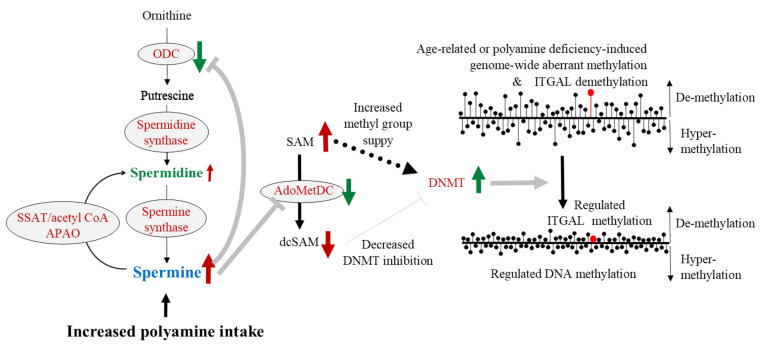
The effects of increased polyamine intake on enzyme activities and substance levels related to polyamine metabolism and gene methylation.

**Figure 5 cells-11-00164-f005:**
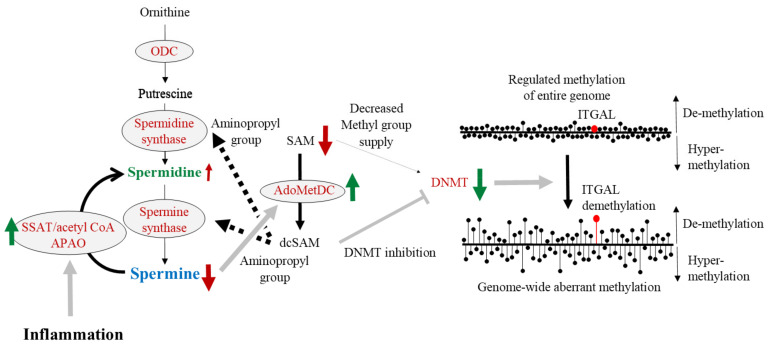
The effects of chronic inflammation on enzyme activities and substance levels related to polyamine metabolism and gene methylation.

**Figure 6 cells-11-00164-f006:**
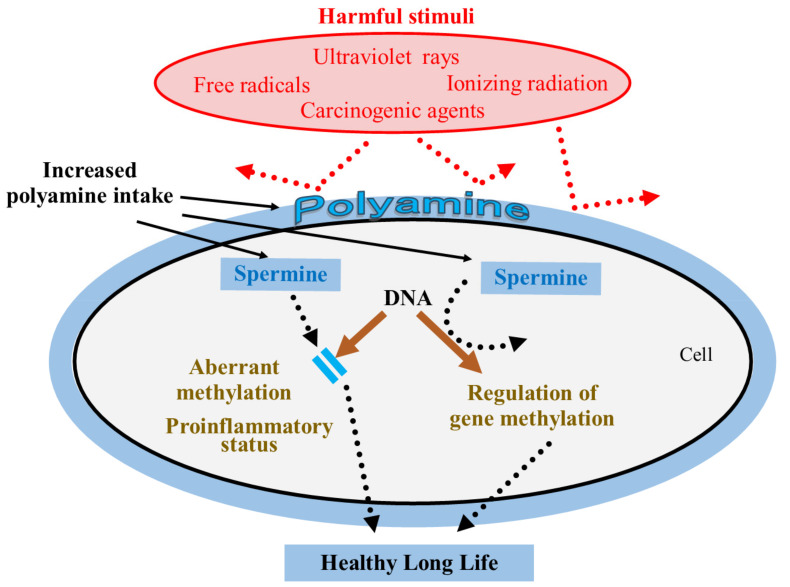
Bioactivities and mechanism of polyamines contributing to healthy long life.

## Data Availability

No new data were created or analyzed in this study. Data sharing is not applicable to this article.

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
