# Peer review of "Overview of Polyamines as Nutrients for Human Healthy Long Life and Effect of Increased Polyamine Intake on DNA Methylation"

_cells, 2022, doi:10.3390/cells11010164_

Round 1
Reviewer 1 Report
This review on the role of nutrient polyamines in longevity is very informative and provides both a comprehensive overview of the state of research in the field and insight into the author's own work. I have several suggestions:
- Please mention what criteria were used to select articles for inclusion in the review.
- I suggest including a section or statement on future research directions in the field.
- If possible, please provide some practical recommendations regarding the inclusion of polyamines as nutrients in the diet. If this is not possible at this stage, please mention that.
Author Response
- Please mention what criteria were used to select articles for inclusion in the review.
Reply: There is no time period in which the papers were published or keywords used to select the papers. However, I try to adopt both conflicting research results as much as possible before discussing them. In doing so, I do not take the so-called impact factor into consideration at all.
- I suggest including a section or statement on future research directions in the field.
Reply: Thank you very much for your kind suggestion. There are many problems to be solved and future perspective, so this will be a long document. Therefore, I do not wish to include an additional summary of future perspectives in this section. The current problems to be solved are described in each section (subheading).
- If possible, please provide some practical recommendations regarding the inclusion of polyamines as nutrients in the diet. If this is not possible at this stage, please mention that.
Reply: At this point, all I can say is that we should try to consume foods with high polyamine content. In the future, it may be possible to create a new healthy longevity diet by mixing synthetic polyamines into foods with low polyamine concentrations. One of the problems of regional disparities in health and longevity is the difference in the polyamine content of the foods consumed in each region, and I believe that correcting this problem by adding synthetic polyamine will help expand the number of regions with healthy longevity.
Reviewer 2 Report
Overview of Polyamines as Nutrients for Human Healthy Long Life and Effect of Increased Polyamine Intake on Gene Methylation” By Soda carefully describes the state of the art about Polyamines, spermidine and spermine metabolism and their role in methylation and healthy longevity.
My suggestion to the author is to reduce, or even delete, the paragraphs about Gene methylation and Aging, Controversial on Autophagy and Life Span, Gene methylation and Aging, which are out of the scope of this review. Rather, I would shortly describe the content of the paragraphs when touching the arguments in the context of polyamine. English style needs to be improved.
Author Response
- My suggestion to the author is to reduce, or even delete, the paragraphs about Controversial on Autophagy and Life Span and Gene methylation and Aging, which are out of the scope of this review.
Reply: I deleted the paragraphs in Sentences in “7. Controversial on autophagy and life span” significantly, and the sentences I left were moved to the end of “6. Biological activities of polyamine” in the revised version. The subheading of “7. Controversial on autophagy and life span” was deleted in the revised version.
The subheading of “8. Aging and inflammation” was changed to “7. Aging, proinflammatory status, and DNA methylation” in the revised version. I reduced paragraphs in the “7. Aging, proinflammatory status, and DNA methylation” in the revised version.
Sentences in “9. Gene methylation and aging” were significantly deleted, and I deleted the subheading of “9. Gene methylation and aging”. And the part of sentences in “9. Gene methylation and aging” were moved to the end of “7. Aging, proinflammatory status, and DNA methylation” in the revised version.
- Rather, I would shortly describe the content of the paragraphs when touching the arguments in the context of polyamine.
Reply: Thank you very much for your suggestion. I deleted unnecessary paragraphs.
- English style needs to be improved.
Reply: I submit revised version without further English editing by native speaker because I did not have enough time to resubmit my revised version. If you feel content of my review article is suitable for publication, I will ask the editor to give me some time to proofread the document.
Reviewer 3 Report
Although the review is quite (indeed too) extensive, the central core of the review concerns the emphasis of the author's papers on LFA-1 and spermine. Regarding the emphasis on one's papers, there is a questionable sentence at the end of the review that the referee does not consider appropriate for a scientific paper (And, after a long period 801 of non-responsive and silent review of a submitted paper, we could publish for the first 802 time the effect of increased polyamine intake on the lifespan of mice [52]).
LFA-1 is an adhesion molecule that plays a key role in leukocyte infiltration and therefore in the inflammatory process. In the inflammatory process underlying inflammaging, the recruitment of monocytes plays a marginal role, if any. Others are the more important causes of this chronic low-grade inflammation.
In biomedical terminology senescence is not synonymous with aging, it is in common language and in literary language.
Author Response
- Although the review is quite (indeed too) extensive, the central core of the review concerns the emphasis of the author's papers on LFA-1 and spermine. Regarding the emphasis on one's papers, there is a questionable sentence at the end of the review that the referee does not consider appropriate for a scientific paper (And, after a long period 801 of non-responsive and silent review of a submitted paper, we could publish for the first 802 time the effect of increased polyamine intake on the lifespan of mice [52]).
Reply: I deleted sentences extensively. The last part of my article poses a problem existing in the scientific community. Several scientists I trust have read it and pointed out that these are rather very important sentences. And, several scientists told me that they had similar experience that I have suffered. Additionally, editor and reviewers other than you had no comments. Therefore, I consider it unnecessary to delete it, rather should be described. We in the small group of scientists are very happy if you understand our situation.
- LFA-1 is an adhesion molecule that plays a key role in leukocyte infiltration and therefore in the inflammatory process. In the inflammatory process underlying inflammaging, the recruitment of monocytes plays a marginal role, if any. Others are the more important causes of this chronic low-grade inflammation.
Reply: Inflammation has a complex mechanism and a variety of triggers. There is no question, however, that this is an article describing the bioactivity of polyamines and their contribution to lifespan extension, not a description of the mechanisms of inflamm-aging.
- In biomedical terminology senescence is not synonymous with aging, it is in common language and in literary language.
Reply: Thank you very much about “senescence”. I replaced “senescence” in line 167 by “old age”. And “aging” in line 525 (469 in revised ver.), 551 (475 in revised ver.), and 656 (584 in revised ver.) is replaced by “senescence”. Sentence in line 567 was deleted in revised version.
Round 2
Reviewer 2 Report
I have no further comments
Reviewer 3 Report
The author replies almost satisfactorily to my comments